# Semiochemicals from Domestic Cat Urine and Feces Reduce Use of Scratching Surfaces

**DOI:** 10.3390/ani14030520

**Published:** 2024-02-05

**Authors:** Lingna Zhang, Edgar O. Aviles-Rosa, Zhaowei Bian, Kaz Surowiec, John J. McGlone

**Affiliations:** 1Laboratory of Companion Animal Science, College of Animal Science, South China Agricultural University, Guangzhou 510642, China; bzw02052021@163.com; 2Animal & Food Sciences Department, Texas Tech University, Lubbock, TX 79409, USA; edgar.aviles-rosa@ttu.edu; 3Department of Chemistry and Biochemistry, College of Arts and Sciences, Texas Tech University, 1204 Boston Ave, Lubbock, TX 79409, USA; kaz.surowiec@ttu.edu

**Keywords:** cat, urine, feces, semiochemical, scratching

## Abstract

**Simple Summary:**

This study aimed to identify the major volatile compounds from cat urine and feces that differed between intact males and females and evaluate these molecules on cat scratching behavior. Results indicated that males had more 3-Mercapto-3-Methyl Butanol (MMB) in the urine and butanoic acid in the feces than females. And the mixture of MMB and butanoic acid had the potential to deter inappropriate scratching behavior in cats, which provided a new strategy for modifying feline destructive behavior in the home.

**Abstract:**

Scratching is a natural behavior in cats but can cause damage to household furnishings. In this work, we sought to identify potential semiochemicals in the urine and feces of domestic cats that may modify cat scratching behavior. Sex differences among adult, intact cats were examined for volatile molecules in their urine (*n* = 7 females, 7 males) and feces (*n* = 8 females, 10 males) using gas chromatography-mass spectrometry (GC-MS). Males had seven times more 3-Mercapto-3-Methyl Butanol (MMB, *p* < 0.001) in the urine and 98% more butanoic acid (*p* = 0.02) in the feces than females. One mL of mineral oil without (i.e., control) or with MMB (0.1 µg/mL) and butanoic acid (100 µg/mL; i.e., treatment), which corresponds to the estimated biological amount in a single elimination from a male cat, were evaluated for their effectiveness in modifying the use of scratching devices by cats. Two identical cardboard standing scratchers, treated with either the control or the solution containing both semiochemicals delivered through a hanging cotton sock were placed side by side in a home/shelter environment. The preference test consisted of exposing individual cats (*n* = 28) to both scratchers for 20 min and recording the duration and frequency they interacted or scratched each scratcher. The semiochemical solution significantly decreased scratching time (21.19 ± 3.8 vs. 6.08 ± 3.8 s; *p* < 0.001) and interaction time (31.54 ± 5.9 vs. 12.90 ± 5.9 s; *p* = 0.0001) and tended to reduce scratching frequency (1.49 ± 0.3 vs. 0.82 ± 0.3 times; *p* = 0.07) compared with the control solution. The male-representative solution of MMB and butanoic acid was aversive to cats and might have future applications in protecting furniture from the destructive scratching or in modifying behavior of domestic cats.

## 1. Introduction

Cats rely heavily on olfactory and chemical communications for individual identification, territory/route marking, and reproductive recognition and as an alarm mechanism [1,2,3]. Semiochemicals include any chemical signal given off by one individual that provides a message, altering the physiology and/or behavior of another individual, while pheromones are a type of semiochemicals for communication within the same species [4]. In the current study, we will also use semiochemical to describe those chemical communicative compounds, the exact biological functions of which have not yet been completely understood. Candidate pheromones have been reported in cats, with facial pheromones, mammary appeasing pheromone, and pedal interdigital semiochemicals being applied to solve behavioral problems, such as aggression in multi-cat households, urine marking, and anxiety in a novel environment [5,6]. Less work has been carried out on the identification and application of potential pheromones/semiochemicals from cat urine and feces, even though odor signals from cat eliminations may convey individual and/or sex information to conspecifics. For example, intact male cats and estrous female cats exhibit urine marking on vertical surfaces [7]. Feces were found unburied at the peripheral but not the core areas of their home range [8,9]. Felinine and its volatile derivatives, including 3-mercapto-3-methyl butanol (MMB) and 3-mercapto-3-methylbutyl formate, can be detected in the urine of male cats and some small felids like bobcats and leopards [10]. With a characteristic sulfur odor, MMB is believed to be a male cat sex pheromone because a higher concentration of MMB is found in the urine of mature intact male cats compared to the urine of females or castrated males [10]. A recent study reported that feces from intact male adult cats had higher levels of propanoic acid, 4-methyl-pentanoic acid, and MMB than feces from non-estrus intact females [11]. The same volatile fatty acids were also detected at the perianal area and were suggested to serve the purpose of individual rather than sex identification in cats [11]. Few studies have evaluated cat urinary and fecal components as behavioral modifiers. Organic extracts from the urine of intact male cats induced sniffing and flehmen response in both laboratory male cats and bobcats of both sexes housed in outdoor enclosure [12]. A patent reported that the use of L-felinine in cat litter attracted cats to eliminate in the litter box (U.S. Patent 20160309676A1), but MMB, the major felinine metabolite, did not increase the use of litter box in cats [13].

Scratching, as a natural behavior for cats, is speculated to serve functions of nail polishing, extension of hind limbs, and providing visual and chemical signals for conspecific communications [14,15]. The ability to engage in scratching behavior is not only natural, but vital to the welfare of cats because one way to improve animal welfare is providing opportunities for that animal to engage in natural, species-typical behaviors. However, when cat scratching behavior is exhibited indoors on the furniture, it is often considered problematic by cat owners and referred as inappropriate scratching [16]. A common solution to this problem is to redirect this behavior to scratching devices. Successful scratching redirection depends on multiple factors, such as the scratcher type and its location in the household [17,18,19], individual cat preferences [20,21], and the attractiveness of the scratchers [6]. The use of herbal cat attractants such as catnip and/or silver vine to attract cats and increase playful behavior has been well documented [22]. Catnip and its extracted oil have been reported to induce scratching when placed on scratching devices in adult cats [6] but not in kittens less than 8 weeks old, possibly due to the immature behavioral development in young cats [20]. Application of pheromone/semiochemicals that can attract or deter cats to the provided scratchers might assist in reducing inappropriate scratching. The aim of the current study was to identify and quantify the major volatile compounds from cat urine and feces that differed between intact males and females, and to investigate the effect of these molecules on the use of preferred scratchers in cats.

## 2. Methods and Materials

### 2.1. Animals

All research was approved by the Texas Tech University Institutional Animal Care and Use Committee (Protocol 17010-02) prior to the beginning of the work. Healthy adult cats (≥1 year of age) were recruited from owners and the local shelter (Lubbock Animal Shelter and Adoption Center, Lubbock, TX, USA). Detailed information of the cats is shown in Appendix A Table A1. Cat owners were blind to the treatments. Ten intact male (M) and eight anestrous female (F) cats from the shelter were included for fecal collection. Seven intact males and seven intact females from the shelter were included for urine collection. Cats in the shelter were housed individually in kennels (90 cm length × 65 cm width × 75 cm height) with segregated areas for defecation, feeding, and resting. They were fed the same diet (Hill’s, Science Diet Optimal Care^TM^, premium natural cat food, Chicken Recipe, Hill’s Pet Nutrition, Inc., Topeka, KS, USA) for at least three weeks before sample collection. A total of 28 cats, including 7 M and 7 F from the shelter and 7 neutered male (NM) and 7 spayed female (SF) household cats were included in the behavioral assay. Cats from households were freely kept inside the house and had access all the rooms. Cats from the shelter were not familiar with each other and so were the owned cats except for cats living in the same household.

### 2.2. Sample Collection

Samples were collected in December 2018 and none of the female cats showed signs of being in estrus (e.g., increased frequency of calling, lordosis, and other sexual behaviors). Unscented nonabsorbent litter (Petconfirm, Nancy Ridge Technology Center, San Diego, CA, USA) was used for urine collection. Fecal samples were collected either from the litter box or with the free-catching method. The researcher checked the litterbox every 30 min during the day (from 8:00 a.m. to 6:30 p.m.) for urine and feces. Contaminated samples were discarded. Urine was collected with centrifuge tubes (Conical Centrifuge Tubes, Falcon^®^, Newport, TN, USA) and fecal samples with whirl-pack bags (Sigma-Aldrich, St Louis, MO, USA). Samples were placed on ice immediately after collection, transferred to the lab, and stored at −80 °C until extraction.

### 2.3. Urine Extraction

Urine samples were thawed at room temperature for about 10 min. After thawing, NaCl was added until saturation to precipitate urinary proteins. The urine was then vortexed for 2 min and centrifuged at 3000 rpm, 15 °C for 10 min. The supernatant was filtered using a 0.2 µm cellulose Acetate (CA) syringe filter (Whatman, GE Healthcare, Hatfield, UK) and 1 mL of 200 ppm 4-ethyl phenol (97%; Sigma-Aldrich, USA) in ddH_2_O as an internal standard solution was added to 9 mL of the filtered urine. The 10 mL mix was then vortexed and filtered through a reversed phase HyperSep C18 SPE cartridge (2 g bed weight; 40–60 µm particle size; 60 Å pore size; 15 mL column capacity; Thermo Fisher Scientific, Bellefonte, PA, USA) at a steady rate of 0.25 mL/sec using a 12-valve vacuum manifold (Thermo Fisher Scientific, Bellefonte, PA, USA). The cartridge was previously conditioned with 10 mL of acetonitrile and 10 mL of ddH_2_O. After filtration, the cartridge was washed with 10 mL ddH_2_O, and centrifuged at 3000 rpm, at 15 °C for 3 min to remove remaining water from the column. Molecules were subsequently eluted with 2 mL acetonitrile. To the eluted sample, 0.1 g NaCl and 0.1 g MgSO_4_ were added to separate the aqueous layer and organic layer [23]. After 15 min, the upper layer was carefully transferred to a 2 mL screw-capped Gas Chromatography (GC)-vial for chemical analysis (Thermo Fisher Scientific, Bellefonte, PA, USA).

### 2.4. Feces Extraction

Two grams of thawed (previously frozen) feces was placed in a centrifuge tube with 0.5 µL of heptanoic acid (>99%, Sigma-Aldrich, USA) as the internal standard and 5 mL of acetonitrile, and vortexed for one minute. Samples were centrifuged at 3000 rpm, 15 °C for 10 min. Once centrifuged, the supernatant was filtered with a polytetrafluoroethylene (PTFE) 0.2 µm syringe filter (VWR North America, Denver, CO, USA). Each of 0.1 g NaCl and MgSO_4_ was added to the filtered solution to separate the aqueous layer from the organic layer [23]. After 15 min, the upper layer was carefully transferred to a GC-vial for chemical analysis. The litter used for sample collection was also extracted with acetonitrile and analyzed in the same way as feces to confirm that volatiles found in feces were not litter residues.

### 2.5. Gas Chromatography-Mass Spectrometry (GC-MS)

The fecal and urinary extracts were analyzed using GC-MS (Thermo Scientific Trace GC-MS Ultra connected to single quadrupole ISQ MS, Thermo Fisher Scientific Inc., San Jose, CA, USA) that was equipped with an SPB-PUFA capillary column (30 m length × 0.25 mm i.d. × 0.20 μm; Sigma-Aldrich, USA) containing poly alkylene glycol-bonded stationary phase. One microliter of the sample was injected by an auto sampler in splitless mode into the injection port that was pre-heated to 250 °C. Helium was the carrier gas and flowed at 1.2 mL/min. The temperature program of the oven was as follows: 100 °C for 2 min, then increased by 7 °C/min to 220 °C, and then held at 220 °C for 15 min. The temperature of the mass spectrometer ion source was 225 °C during analysis. Mass spectra were recorded in electron-impact (EI) mode at 70 eV with a mass range from 40 to 450 amu. Compounds of interest were initially identified by matching the obtained mass spectra with a reference library in the instrument control software. The identities of peaks of interest were further confirmed by comparing the mass spectra and retention time with analytical standards, including MMB (≥98%, Sigma-Aldrich, USA) and butanoic acid (≥99%, Sigma-Aldrich, USA).

### 2.6. Behavioral Assay

Two socks rinsed either with the control or the volatile-containing solution, were hung on two identical cardboard standing scratchers. Both scratchers were made of a square wood board (44 cm × 44 cm × 1.5 cm) with a cardboard column (13 cm × 77 cm) attached to it (Figure 1). For shelter cats, the experimental room was the feline interaction center at the shelter measured 4.39 m × 3.63 m. The room had a standing bookcase, a table, a cat tree and some toys. The scratchers were placed next to the bookcase and the other items were removed from the nearby testing area. For household cats, the test occurred in one room at owner’s place. The two scratchers were placed side by side in the room and sides were changed between trials. A camera (LINNSE, Camcorder Full HD, Amazon, Seattle, WA, USA) was placed in front of the scratchers to videotape the trials. Cats were introduced to the experimental room one at a time and they were given 20 min to interact with the scratchers. A habituation period was not specified; therefore, the 20 min testing period included the time for the cat to habituate to the room and explore both scratchers. The 20 min test was conducted only once for each cat. The testing area was relatively novel for shelter cats, which is the feline interaction center at the shelter with space for potential adopters to interact with candidate cats. As for owned cats, the test occurred in one of the rooms in the house, therefore, was familiar for the cats. A single trained and validated person watched the videos and recorded how long (duration) and how often (frequency) scratching and interactions exhibited to the scratchers occurred over the 20 min experimental period. Behavioral coding by the same person was validated in a previous study where use of scratchers in cats was studied in detail and high intra-observer agreements were found for these behavioral measurements (Spearman’s rank-order correlation ρ > 0.98) [21]. The videos were watched using fast-forwarding at ×5 times to find video segments of interest. Each short video was then watched in detail with continuous sampling in real time. The definition of scratching and interactions are presented in Table 1. The interactions mentioned in the current study referred to the total interactions, which included scratching and other non-scratching interactions.

### 2.7. Statistical Analysis

All statistical analyses were conducted using SAS 9.4 (SAS Inst., Inc., Cary, NC, USA). Peak areas of volatiles from the chemical analysis were calculated using Qual Browser within Xcalibur (Thermo Fisher Scientific Inc., Waltham, MA, USA). To be included in the final analysis, the molecule was required to be present in all individuals of the same sex and not be presented in the litter used for sample collection. An area ratio for individual peaks was calculated as the peak area of the interested molecule divided by the peak area of the internal standard. Data of peak area ratio were examined for parametric analysis using the Shapiro–Wilks and Levene’s test. Data that met the assumption for normality were analyzed using the student t-test and the degree of freedom was adjusted with Satterthwaite approximation if the assumption of equal variance was not met. Wilcoxon rank sum test was applied to analyze the data when assumption of normality was not met. The sex effect was considered significant at *p* ≤ 0.05. Calibration curves were made by running analytical standard at different concentrations using the same GC-MS protocol [25]. The equation of the curve was further used to estimate the fecal or urinary concentrations of the analytes of interest.

For the behavioral data, assumptions of parametrical analyses were not met based on the Shapiro–Wilks test and Levene’s test and were analyzed with the Wilcoxon signed rank test. Preference index (PI) data, calculated as described in Table 1, were transformed using the arcsine square root transformation and analyzed as repeated measures using GLIMMIX. The model included cat sex and treatment (TRT) and the interaction between sex and treatment (sex*TRT) with cat as random effect. Significant differences were considered at *p* ≤ 0.05 and a tendency at 0.05 < *p* ≤ 0.10.

## 3. Results

### 3.1. Urine Volatiles

Six volatiles were identified in the urine of both intact male and female cats (Figure 2, Table 2). When the peak area ratio of these volatiles was compared between sex, male cats had higher 3-mercapto-3-methyl butanol (MMB, *p* = 0.02) and 4-heptanol, 2, 6-dimethyl (*p* = 0.05) in their urine than females (Table 3).

### 3.2. Feces Volatiles

Twenty major volatiles were identified in the feces of both intact male and female cats (Figure 3, Table 4). When the peak area ratio of these volatiles was compared between sexes, male cats had higher butanoic acid (*p* = 0.04) and lower 9-octadecenoic acid (*Z*)-ethyl ester (*p* = 0.03) in their feces compared to female cats (Table 5).

Of the four volatiles differed in the urine and feces between male and female cats, MMB (the recognized male sex marking pheromone) and butanoic acid (abundant in the feces) were selected for further investigation. The other two molecules were not selected because their identities were not able to be confirmed using analytical standards. Specifically, standard MMB was dissolved in acetonitrile to make an original standard solution in 500 ppm, then diluted to 3.9 ppm with twofold serial dilution method. The standard solution of butanoic acid ranged from 5000 ppm to 62.5 ppm in acetonitrile, also using twofold serial dilutions. The calibration curve for MMB and butanoic acid were built with the concentrations of different standard solution and their corresponding peak areas (R^2^ ≥ 0.99). Content of MMB and butanoic acid (corrected for the fecal dry matter content) in the urinary and fecal samples were estimated with calibration curves (Table 6). Both MMB and butanoic acid were at higher (*p* < 0.05) concentrations in male samples than in female samples (Table 6).

### 3.3. Behavioral Assay

The estimated amounts semiochemicals from one elimination of urine and feces of an intact male cat (i.e., approximately 15 mL urine and 5 g dry matter feces) were dissolved in one milliliter of mineral oil (i.e., 0.1 µg/mL of MMB and 100 µg/mL of butanoic acid) and used as the treatment solution for the behavioral assay. The control solution was one milliliter of mineral oil without volatiles added.

Cats preferred (*p* ≤ 0.07) the control scratcher over the volatile-treated scratcher when main treatment effect was tested using Wilcoxon sighed rank test. Longer duration of scratching (21.19 ± 3.8 s vs. 6.08 ± 3.8 s; *p* < 0.0001) and interaction (31.54 ± 5.9 s vs. 12.90 ± 5.9 s; *p* = 0.0001) and a trend of higher scratching frequency (1.49 ± 0.3 times vs. 0.82 ± 0.3 times; *p* = 0.07) were observed during the 20min period for the control scratcher compared to the volatile-treated scratcher (Figure 4).

No sex effect was observed for preference index (PI)s of all behavioral measurements (i.e., duration and frequency of scratching and interaction). Focusing on the main treatment effect, PIs for scratching duration and frequency of the control scratcher were higher (*p* < 0.05) than those with the treated scratcher (Table 7), indicating a preference for the control scratcher. The treatment by sex interaction was observed (*p* = 0.03) for the PI of interaction duration, being significantly higher for the control scratcher than the treated scratcher in intact (F, 0.93 vs. 0.07; M, 0,76 vs. 0.24 for control and treatment, respectively) but not among castrated cats (*p* > 0.10; Figure 5).

## 4. Discussion

We found fewer volatiles in the urine samples compared to what has been reported by others [10,12,26]. However, software allowed us to subtract male from female data and show the few differences between the sexes. Difference in the methodologies might also contribute to this inconsistency. The extraction method used for feces did not work out for urine samples, and the step of filtration through a reversed phase HyperSep C18 SPE cartridge was added to urine extraction. We identified and quantified 3-Mercapto-3-Methyl Butanol (MMB) and p-cresol (4-Methyl-Phenol). The relative abundance of MMB compared between male and female cats (i.e., approximately 1 to 8) agreed with those reported by Miyazaki et al. (2006) [10]. Few studies have evaluated the concentration of MMB in the cat urine, but MMB was described as having a typical male cat odor at concentrations of 0.01–1 ppm [27], significantly lower than the MMB estimated in male and female cats in this study. All the samples were collected and transferred to the laboratory on ice and stored at −80 °C for no more than a week until analysis. The elapsed time (~4 h) between the shelter and laboratory could contribute to the higher MMB concentrations. Cat urine develops the typical odor peaking in intensity by ~12 to 24 h, which is a result of MMB being decomposed from felinine, a urinary sulphur amino acid by microbial activity and/or oxidation by the air [10,27]. The excretion rate of felinine in the urine was reported to be 95 mg/day in intact male cats, which was higher than intact females (19 mg/day) [28]. The high excretion rate of felinine requires higher intake of dietary sulphur amino acids (e.g., cysteine). Therefore, MMB as the main derivative of felinine may serve as an “honest signal” of hunting skills in the urine of male cats since muscle meat is the main resource for cysteine [28].

Major volatiles identified in cat feces by the current study (e.g., short chain fatty acids, P-cresol, indole) were also reported by Miyazaki et al. (2018) [11]. MMB was also identified in the cat feces and in higher concentration in intact males (50 ng/g wet feces) than females [11]. But we could not detect MMB from the feces. Miyazaki et al. (2018) also reported more propanoic acid and 4-methyl pentanoic acid existing in male feces than female feces. Our results disagreed with them in that butanoic acid was found to be 74.5% higher in male feces than in female feces and other fatty acid did not differ between sexes. Cats were able to distinguish between two sets of mixed fatty acids in different ratios that mimic the compositions of fatty acids in the feces of two individual males [11]. This evidence suggests to us that fatty acids may be involved in individual and sexual identification in cats. A role of fatty acids in olfactory communication is also seen with the cat facial and mammary pheromones, both containing a variety of volatile fatty acids [5]. The concentrations of butanoic acid reported in our cat feces were slightly higher but close to what others have reported [29,30], possibly due to a higher starch and fermentable fiber content in the diet for our cats [29,31]. Future studies are needed to better understand the interaction between sex and diet on the volatile fatty acids in cat feces.

The information conveyed by urine spraying in cats is not completely understood but has been suggested to involve sexual and identity communication, territory/route marking, and stress [32,33]. Flehmen response is often induced in a cat after it sniffs cat urine, especially the fresh ones from unfamiliar intact males [12]. Urinary extracts of male cats were shown to attract cats and bobcats, but deterred them from urination and defecation in the testing area [12]. Fecal marking is barely studied in cats. Heavier male cats tend to bury their feces closer to the core area than lighter males, indicating that defecating behaviors in male cats may reveal information of social ranking [8]. Only a few studies have investigated the application of specific urinary or fecal volatiles in modifying cat behaviors, possibly due to their pungent odor. We reported here the application of MMB and butanoic acid reduced scratching and total interactions exhibited to the scratcher in cats. None of the cats in the present study exhibited a flehmen response during the behavioral assay even though they did sniff both treatments. Miyazaki et al. (2016) reported that both male and female cats showed interests in MMB but not to the pure felinine, but information about the dosage of MMB used and whether MMB induced flehmen behavior or just increased sniffing in those cats was not detailed. One study reported that MMB added to the litter at 50 μg/kg of litter did not alter the use of litterbox in cats [13]. An explanation for the lack of flehmen response in our cats to the solution containing MMB and butanoic acid is that MMB alone was not enough for representing the urine since urine extracts usually contain MMB and other sulphur-containing volatiles [10]. Alternatively, interaction between butanoic acid and MMB blocked the flehmen response of MMB. Because MMB alone was neutral or attractive to cats, we did not test MMB alone, rather we attempted to reconstitute the active molecules that are unique to male cat excretions. Testing MMB and butanoic acid separately in the future may verify this latter hypothesis. The role of MMB in contexts other than sexual and territorial communication also requires further study. In addition, most cats in the current study finished interacting with both scratchers within the first 10 min of the testing period, therefore suggesting shorter assessments possibly be utilized in the future to make trials easier for both cats and researchers.

The treatment solution with MMB and butanoic acid was assumed to represent a male odor in cats. Cats scratch in the area they live and along the daily pathway to mark a looser home range [7], after sleeping [24], and near feeding and defecation area [5]. Free-roaming cats scratch more often in the presence of other cats [34], but they do not tend to over-mark urinary and fecal marks of others [12]. Cats may also prefer to scratch on the control scratcher nearby but not right on the scratcher treated with male-representative volatiles. Alternatively, cats may avoid the treated-scratcher to avoid confrontation with other cats and a recent hunting area because free-roaming cats whether live a solitary life or form highly gregarious social colonies with one another [35], use urinary and fecal marks to separate themselves apart temporally and spatially during hunting and social communication [34]. Differed responses to the treatment solution between intact and castrated cats were observed, in that interaction duration was reduced in intact but not in castrated cats. Little literature is available for explaining the result. Intact males and anestrous females might show avoidance to the male-representative volatiles to reduce conflicts from confronting a male cat. Castrated animals might exhibit less avoidance due to a lack of reproductive motivation and the hormonal change after gonadectomy [36]. Sex hormones can affect the olfactory sensitivity to multiple odorants [37]. Alternatively, environmental and dietary differences between the two groups of cats may have also played a role since the factors of shelter versus home-reared are completely confounded. A relatively impoverished environment may alter the stress hormones [38] and body chemicals in shelter cats. The castrated cats were tested in homes, at their owner’s place, and the intact cats in the play room at the shelter. Shelter cats might be more vigilant and less explorative at the play room as it is a novel environment [39]. This is less possible because the total interaction frequency with both scratchers were not much different between shelter and pet cats (5.36 ± 0.37 times vs. 3.79 ± 2.12 times, *p* = 0.32; Wilcoxon rank sum test). In contrast, pet cats exhibited less interaction in duration with both scratchers compared to cats in the shelter (69.1 ± 70.3 s vs. 21.5 ± 17.8 s; Wilcoxon rank sum test). In addition, no significant sex effects were observed on other behavioral measures (i.e., scratching frequency and duration). During the test, apparent stressful behaviors (e.g., hiding, escape attempts, agonistic vocalization) were not observed for both shelter and owned cats as well [40]. Urinary and fecal samples were collected from shelter cats put on the same diet, and concentrations of chemicals in the treatment solution were determined based on their contents in the samples from shelter male cats. The pet cats in our study were not standardized for their diet, imposing diet as a potential factor for the behavioral difference between shelter and pet cats. In the future, it would be interesting to study how diets with varied ingredients affect urinary and fecal volatiles in cats and if volatiles mimicking body chemicals from cats on one diet will cause behavioral differences in these cats and cats on another diet. Meanwhile, the treatment solution tested in the current study was reported as detectable and aversive upon approaching (<0.5 m) by most owners. Therefore, it would be useful to investigate the minimal concentration of treatment solution that are unnoticeable or acceptable to humans while still protect furniture and other objects from being scratched. From evolutionary considerations, feline social rank and testosterone levels might correlate with the concentration of the semiochemicals in the urine and feces of intact males [10,27]. Future studies could explore the correlation between these variables and test out the intraspecific effects of different combination of candidate volatiles from cat urine and feces in various concentrations.

## 5. Conclusions

In summary, intact male cats had higher MMB and butanoic acid in their urine and feces compared to intact females. The solution containing the estimated amounts of MMB and butanoic acid from one elimination of a male cat had aversive effects on the use of scratchers, especially in intact cats. In the future, it will be worthwhile to test the two volatiles separately and investigate their potential behavior-modifying effects in other contexts (e.g., use of litter box). The mixed solution of MMB and butanoic acid may have an application in protecting furniture or other objects or areas from being scratched by cats, which has the potential to inform recommendations that could increase cat welfare and improve human–cat relationships. For example, the semiochemical solution can be applied to a cloth and put over the area where the owner wants to decrease inappropriate scratching (e.g., side of the couch). At the same time, the owner can positively reinforce use of the cat scratcher to redirect the cat’s scratching behavior (e.g., scratcher with catnip/silver vine) [21].

## Figures and Tables

**Figure 1 animals-14-00520-f001:**
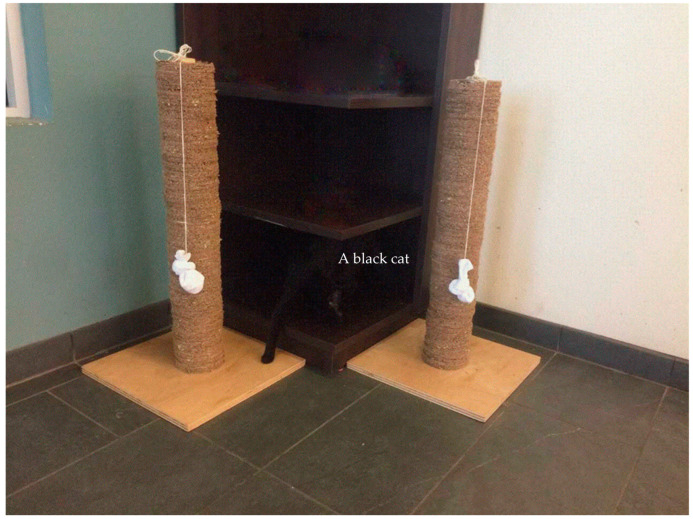
An example of two cardboard standing scratchers with hanging socks treated control or volatile-containing solution, respectively. The scratchers had a square wood base that measured 44 cm × 44 cm × 1.5 cm and an attached round standing column (77 cm in height × 13 cm in diameter) that contained a center square wood stick (5 cm × 5 cm) covered with round stacked cardboard. One centimeter height of the cardboard column contained four pieces of the stacked cardboard.

**Figure 2 animals-14-00520-f002:**
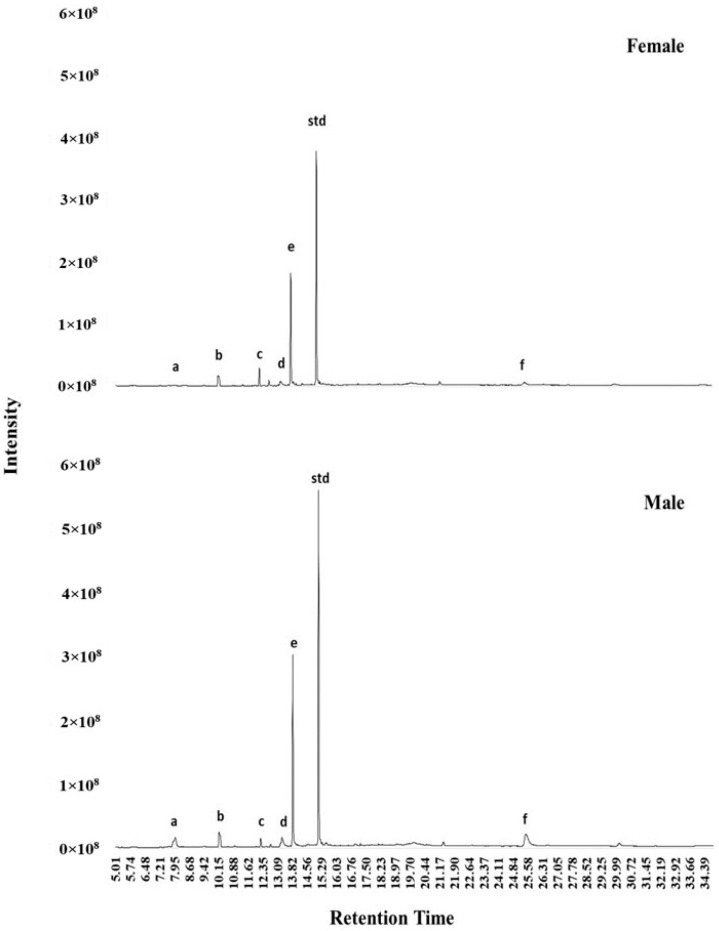
Representative chromatogram of volatiles in the urine of male and female cats. Letters a–f correspond to the candidate molecules in Table 2. Std: standard.

**Figure 3 animals-14-00520-f003:**
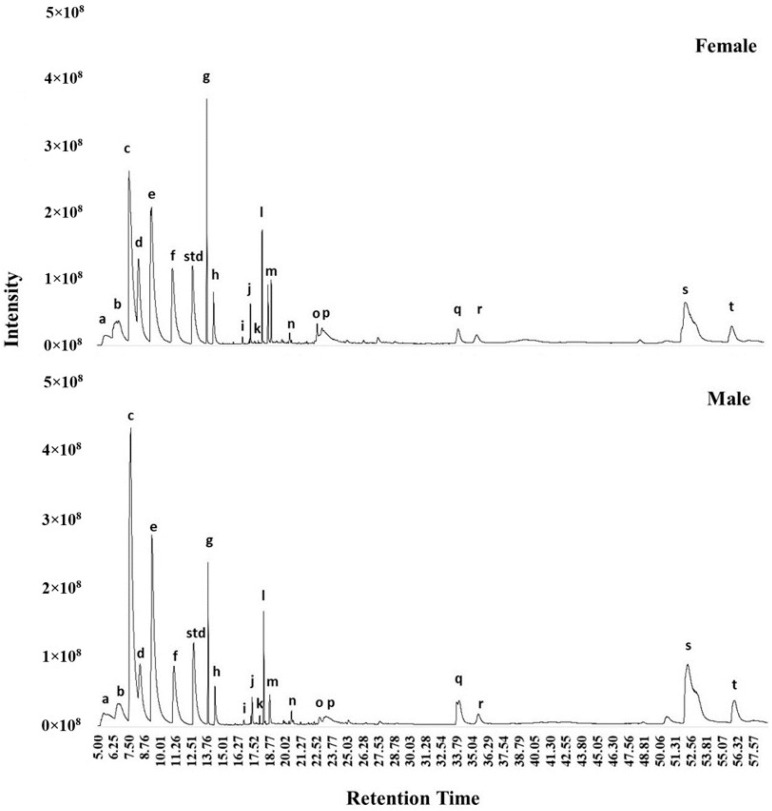
Representative chromatogram of volatiles in the feces of female and male cats. Letters a–t correspond to the candidate molecules in Table 4. Std: standard.

**Figure 4 animals-14-00520-f004:**
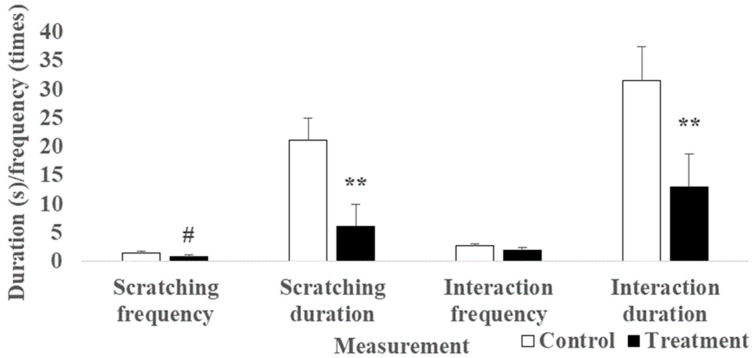
Main treatment effect of treated solution, mix of butanoic acid, and 3-Mercapto-3-Methyl Butanol (MMB) dissolved in mineral oil, compared to control solution (mineral oil) on the behavioral measurements obtained from the standing cardboard scratchers in adult cats (*n* = 28 cats). ^#,^** Least squares mean differed between treatment groups at *p* ≤ 0.10 and *p* ≤ 0.01 based on the Wilcoxon signed rank test.

**Figure 5 animals-14-00520-f005:**
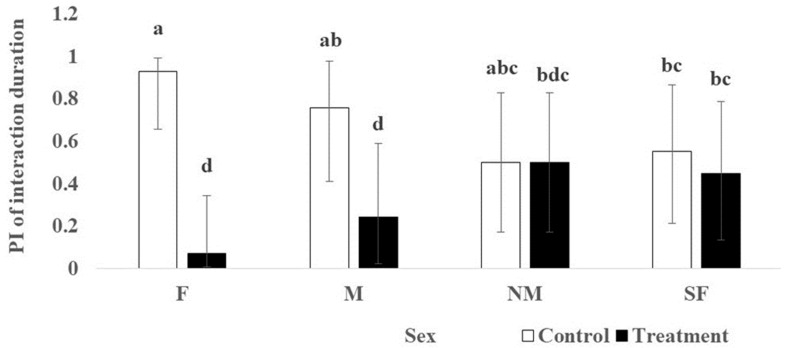
The effects of the interaction between sex, female (F, *n* = 7), male (M, *n* = 7), spayed female (SF, *n* = 7), and neutered male (NM, *n* = 7), and treatment (i.e., control versus volatile-treated standing cardboard scratcher) on the back-transformed preference index (PI) of interaction duration from arcsine square root transformation. The error bars indicated 95% confidence interval for the back-transformed data. ^a,b,c,d^ Least squares mean of arcsine square root transformed data differed within each measurement with different letters.

**Table 1 animals-14-00520-t001:** Behavioral measures exhibited to scratchers by cats [24].

Behavior	Definition
Scratching	With front claws extended, cat grips the material, and its claws withdraw and extend alternately.
Interaction -not scratching	Other active behaviors exhibited by cats on the scratcher and the sock, including climbing on, pawing, kicking or rubbing against the scratcher; sniffing, licking, pawing, or rubbing against the sock.
Total interaction	Sum of scratching and non-scratching interactions.
Preference Index (PI)	PI of specific measurement was calculated as the measurement of one scratcher divided by the measurement summed for both control and treated scratchers.

**Table 2 animals-14-00520-t002:** Major volatiles in the urine of cats.

Peak ^a^	RT	Candidate	Formula	*m*/*z* ^b^	Matching ^c^
a	7.92	3-Mercapto-3-Methyl Butanol *	C_5_H_12_OS	69	92.15%
b	10.17	2-4-Dimethyl-Benzaldehyde	C_9_H_10_O	133	23.48%
c	12.24	Cis-Jasmone	C_11_H_16_O	79	43.04%
d	13.33	2,3-Dethydropiperidin-6-one	C_5_H_7_NO	97	49.14%
e	13.82	P-Cresol/4-Methyl-Phenol *	C_7_H_8_O	107	31.30%
f	25.64	4-Heptanol, 2, 6-Dimethyl	C_9_H_20_O	69	35.85%

^a^ Letter match with peaks in Figure 2. RT: retention time. ^b^ Based on peak of molecular mass spectrum. ^c^ Matching rate between obtained mass spectrum and NIST library database. * Volatile confirmed with analytical standard.

**Table 3 animals-14-00520-t003:** Peak area ratio of volatiles in the urine of intact male and female cats.

Candidate Molecule	Female	Male	SE ^a^	Df ^b^	Statistics ^c^	*p*-Value ^d^
3-Mercapto-3-Methyl Butanol	0.02	0.14	0.02	6	−3.04	0.02
2-4-Dimethyl-Benzaldehyde	0.11	0.09	0.01	12	0.92	0.38
Cis-Jasmone	0.09	0.04	0.02	7	1.42	0.19
2,3-Dethydropiperidin-6-one	0.06	0.08	0.02	12	−0.48	0.64
P-Cresol/4-Methyl-Phenol	0.59	0.65	0.15	8	−0.27	0.79
4-Heptanol, 2, 6-Dimethyl	0.06	0.24	0.04	7	−2.43	0.05

^a^ SE, standard error of least square means. ^b^ Df, degree of freedom (Satterthwaite approximation was applied when data were not homoscedastic). ^c,d^ Test statistics and the significant level of sex effect based on the student *t*-test (*n* = 7 females, *n* = 7 males).

**Table 4 animals-14-00520-t004:** Major volatiles in the feces of cats.

Peak ^a^	RT	Candidate Molecules	Formula	*m*/*z* ^b^	Matching ^c^
a	5.50	Acetic acid *	C_2_H_4_O_2_	60	57.89%
b	6.50	Propanoic acid	C_3_H_6_O_2_	74	34.47%
c	7.59	Butanoic acid/butyric acid *	C_4_H_8_O_2_	60	78.24%
d	8.35	Isovaleric acid/3-methyl-butanoic acid *	C_5_H_10_O_2_	60	69.56%
e	9.32	Pentanoic acid	C_5_H_10_O_2_	60	46.37%
f	11.13	Hexonoic acid	C_6_H_12_O_2_	60	80.65%
g	13.82	P-cresol/4-methyl-phenol *	C_7_H_8_O	107	30.73%
h	14.39	2-Piperidinone	C_5_H_9_NO	99	76.51%
i	16.71	4-Methyl-5-thiazoleethanol	C_6_H_9_NOS	112	67.16%
j	17.27	Hexadecanoic acid,ethyl ester	C_18_H_36_O_2_	88	22.48%
k	17.98	Q-Docecalactone	C_12_H_22_O	85	40.66%
l	18.30	1-H indole	C_8_H_7_N	117	31.36%
m	18.77	Hexadecen-1-ol, trans-9-	C_16_H_32_O	55	14.35%
n	20.56	9-Octadecenoic acid (Z)- ethyl ester	C_20_H_38_O_2_	55	14.97%
o	22.73	Carbonic acid, ethyl octadecyl ester	C_21_H_42_O_3_	91	17.64%
p	23.11	Propanedioic acid, phenol	C_9_H_8_O_4_	91	35.60%
q	34.11	Hexadecanoic acid	C_16_H_32_O_2_	55	14.35%
r	35.62	Oleic acid/8-Octadecenoc acid	C_18_H_34_O_2_	55	17.54%
s	52.39	cis-Vaccenic acid	C_18_H_34_O_2_	55	16.37%
t	56.24	Lenoelaidic acid	C_18_H_32_O_2_	67	10.89%

^a^ Letter match peaks in Figure 3. RT = retention time. ^b^ Based on peak of molecular mass spectrum. ^c^ Matching between obtained mass spectrum and NIST library database. * Volatile confirmed with analytical standard.

**Table 5 animals-14-00520-t005:** Peak area ratio of volatiles of feces in intact male and female cats.

Candidate Molecule	Female	Male	SE ^a^	Statistics ^b^	*p*-Value ^c^
Acetic acid *	0.40	0.52	0.07	−1.07	0.14
Propanoic acid	0.66	0.56	0.10	0.70	0.49
Butanoic acid/butyric acid	2.78	4.85	0.63	−2.28	0.04
Isovaleric acid/3-methyl-butanoic acid	0.93	0.68	0.10	1.76	0.10
Pentanoic acid	2.40	2.81	0.43	−0.61	0.55
Hexonoic acid	1.22	0.93	0.21	0.98	0.34
P-cresol/4-methyl-phenol *	0.53	0.33	0.12	1.69	0.09
2-Piperidinone *	0.25	0.20	0.05	0.62	0.53
4-Methyl-5-thiazoleethanol	0.02	0.02	0.00	0.86	0.40
Hexadecanoic acid, ethyl ester *	0.09	0.08	0.03	0.36	0.72
Q-Docecalactone *	0.01	0.02	0.00	−1.95	0.05
1-H indole *	0.24	0.22	0.07	0.27	0.79
Hexadecen-1-ol, trans-9- *	0.16	0.12	0.04	0.89	0.37
9-Octadecenoic acid (*Z*)-athyl ester *	0.13	0.04	0.03	2.22	0.03
Carbonic acid, ethyl octadecyl ester *	0.31	0.29	0.01	0.20	0.84
Propanedioic acid, phenol *	0.23	0.44	0.11	−0.71	0.48
Hexadecanoic acid	0.15	0.15	0.03	0.02	0.99
Oleic acid/8-Octadecenoc acid *	1.76	2.41	0.43	−1.16	0.25
cis-Vaccenic acid *	0.40	0.50	0.14	−0.62	0.53
Lenoelaidic acid *	0.40	0.52	0.07	−1.07	0.14

^a^ SE, standard error of least square means. ^b,c^ Statistics and significance level of sex difference *(n* = 8 females, *n* = 10 males) based on the student *t*-test or Wilcoxon rank sum test. * Data do not meet the assumption for normality and the Wilcoxon rank sum test was used.

**Table 6 animals-14-00520-t006:** Estimated concentrations of candidate molecules in cat urine and feces.

Molecules	Female CI ^a^	Male CI ^a^	Statistics ^b^	*p*-Value ^c^
MMB (µg/mL urine)	0.00–1.63	2.22–10.99	3.18	<0.001
Butanoic Acid (µg/g DM)	5050–18,348	16271–30,090	2.70	0.02

^a^ Lower and upper value of the 95% confidence interval (CI). ^b,c^ Statistics and significance level of sex difference based on the student *t*-test (MMB, *n* = 7 females and 7 males; butanioc acid, *n* = 8 females and 10 males). DM: dry matter; MMB: 3-Mercapto-3-Methyl Butanol.

**Table 7 animals-14-00520-t007:** The effects of volatiles (i.e., butanoic acid and MMB) on the use of scratchers based on preference index of behavioral measurements in adult cats.

Measurement Preference Index (PI) ^a^	Treatment		*p*-Value ^b^	
	Control	Volatile	TRT	TRT*sex
Scratching duration PI	0.63 0.08 ^b^	0.10 0.28 ^a^	0.0007	0.29
(95% CI)	(0.41, 0.82)	(0.01, 0.27)		
Interaction duration PI	0.70 0.12 ^b^	0.30 0.32 ^a^	0.002	0.03
(95% CI)	(0.53, 0.85)	(0.15, 0.47)		
Scratching frequency PI	0.56 0.09 ^b^	0.14 0.27 ^a^	0.006	0.14
(95% CI)	(0.34, 0.77)	(0.03, 0.33)		
Interaction frequency PI	0.61 0.17 ^b^	0.39 0.29 ^a^	0.08	0.08
(95% CI)	(0.43, 0.77)	(0.27, 0.57)		

^a^ Reverse-transformed least squares means and 95% confidence interval (CI) from arcsine square root transformation. ^b^ Significant levels of treatment (TRT) effect and treatment by sex interaction (TRT*sex) on the arcsine square root transformed data (*n* = 28, 7 intact and castrated males and females).

## Data Availability

Data is contained within the article.

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
