# Peer review of "Semiochemicals from Domestic Cat Urine and Feces Reduce Use of Scratching Surfaces"

_animals, 2024, doi:10.3390/ani14030520_

Round 1

Reviewer 1 Report

Comments and Suggestions for Authors

This paper conducted a biochemical analysis of fecal matter from intact male cats and estrous female cats to identify volatile compounds that differ between the sexes. A preference test was conducted to determine whether these molecules influence cat behavior (approaching and scratching). Two scratchers were presented with either the control or the 26 solution containing both semiochemicals delivered through a hanging cotton sock, and the cats' behavior was observed for 20 minutes. The results showed that the frequency of interacting with the treatment condition and scratching decreased, especially in intact/estrous cats. On the other hand, no difference was observed in spayed/neutered cats. The author concluded that these results may be useful for cats that claw in inappropriate places. This experiment, in which specific volatile compounds were identified in feline fecal matter and their repellent effects were confirmed by behavioral testing, deserves recognition. As for points of concern, they are discussed below.

Please describe a more detailed report on the subject information.

Please state the age of the shelter and pet cat. Also, please provide a brief description of the environment in which each animal was kept (e.g., multiple pets or kept individually in cages).

Please state the relationship (familiar or unfamiliar) between the cats used for behavioral testing and those used for fecal collection.

I am an expert in animal behavior and would like to evaluate the behavioral testing part. I do not see big problem regarding the basic procedure, but would like you to add the following points.

Please specify whether the 20-minute test was conducted only once.

Please specify whether the test was conducted in a familiar place for all cats, as was mentioned in the discussion. It is expected that cats behave completely differently in familiar and unfamiliar places.

Shelter cats are intact and estrous and pet cats are neutered and spayed in the experimental individuals in the behavior test. Since the factors of shelter versus home-reared are completely confounded, please also discuss in more detail regarding whether there were any environmental effects.

We would also like to point out the following points for application.

How strong is the odor for humans? In this experiment, the amount of urine was described as the same as one volume of urine of a cat, so I felt that the odor must be quite strong. This may make it difficult to be practical.

I believe that many cats kept in a house generally are spayed or neutered, so if we increase the concentration, will it be effective for them as well?

There is a possibility that presenting the scent of an unknown male animal could cause anxiety and stress to cats. I would say there is fear that this may induce other problematic behaviors. I would like to know how many cats exhibited stressful behaviors during the test.

The following points should also be added to the evolutionary considerations.

Is it possible that feline social rank and testosterone levels correlate with those smells? For example, are MMB and Butanoic Acid from males with higher testosterone levels more repellent? Please discuss this point.

Author Response

This paper conducted a biochemical analysis of fecal matter from intact male cats and estrous female cats to identify volatile compounds that differ between the sexes. A preference test was conducted to determine whether these molecules influence cat behavior (approaching and scratching). Two scratchers were presented with either the control or the 26 solution containing both semiochemicals delivered through a hanging cotton sock, and the cats' behavior was observed for 20 minutes. The results showed that the frequency of interacting with the treatment condition and scratching decreased, especially in intact/estrous cats. On the other hand, no difference was observed in spayed/neutered cats. The author concluded that these results may be useful for cats that claw in inappropriate places. This experiment, in which specific volatile compounds were identified in feline fecal matter and their repellent effects were confirmed by behavioral testing, deserves recognition. As for points of concern, they are discussed below.

  1. Please describe a more detailed report on the subject information.

R: The detailed information of cats included in the study were shown in appendix table. (L106)

  1. Please state the age of the shelter and pet cat. Also, please provide a brief description of the environment in which each animal was kept (e.g., multiple pets or kept individually in cages).

R: The detailed information of cats was shown in appendix table and changes were made to section 2.1. (L95, 103, 106)

  1. Please state the relationship (familiar or unfamiliar) between the cats used for behavioral testing and those used for fecal collection.

R: The detailed information of cats was shown in appendix table and changes were made to section 2.1. (L103, 106) Besides, the treatment solution for behavioral testing was made with purchased commercial chemicals and diluted into the expected concentration, therefore whether the cats for sample collection and testing were familiar with each other might not play a key role here.  

I am an expert in animal behavior and would like to evaluate the behavioral testing part. I do not see big problem regarding the basic procedure, but would like you to add the following points.

  1. Please specify whether the 20-minute test was conducted only once.

R: The 20-minute test was conducted only once for each cat. This information is added to section 2.6 (L178).

  1. Please specify whether the test was conducted in a familiar place for all cats, as was mentioned in the discussion. It is expected that cats behave completely differently in familiar and unfamiliar places.

R: Thanks for the comment and we added “The testing area was relatively novel for shelter cats, which is the feline interaction center at the shelter with space for potential adopters to interact with candidate cats. As for owned cats, the test occurred in one of the rooms in the house, therefore was familiar for the cats.” to section 2.6. (L179-182)

  1. Shelter cats are intact and estrous and pet cats are neutered and spayed in the experimental individuals in the behavior test. Since the factors of shelter versus home-reared are completely confounded, please also discuss in more detail regarding whether there were any environmental effects.

R: We agree with the reviewer. The relevant content was shown in discussion and we add the highlighted part to address our consideration for the potential environmental effects, which we suggest might not be the main reason for the difference. “Alternatively, environmental and dietary differences between the two groups of cats may have also played a role since the factors of shelter versus home-reared are completely confounded. Relatively impoverished environment may alter the stress hormones (Stella and Croney 2013) and body chemicals in shelter cats. The castrated cats were tested in homes, at their owner’s place, and the intact cats in the play room at the shelter. Shelter cats might be more vigilant and less explorative at the play room as it is a novel environment (Ellis 2009). This is less possible because the total interaction frequency with both scratchers were not much different between shelter and pet cats (5.36 ± 0.37 times vs 3.79 ± 2.12 times, P = 0.32; Wilcoxon rank sum test). In contrast, pet cats exhibited less interaction in duration with both scratchers compared to cats in the shelter (69.1 ± 70.3 s vs 21.5 ± 17.8 s; Wilcoxon rank sum test). Besides, no significant sex effect was observed on other behavioral measures (i.e., scratching frequency and duration). During the test, apparent stressful behaviors were not observed for both shelter and owned cats as well.” (L397-410)

  1. We would also like to point out the following points for application.

How strong is the odor for humans? In this experiment, the amount of urine was described as the same as one volume of urine of a cat, so I felt that the odor must be quite strong. This may make it difficult to be practical.

R: We agree with the reviewer. We were not able to confirm the minimal concentration of treatment solution that can be unnoticeable or acceptable to humans while still exert deterring effect in the current study. This shortcoming is acknowledged and added to discussion. (L418-421) Future studies may provide answers to the question of whether we can reduce the concentration of the mixed solution while still maintaining its deterring effect.

  1. I believe that many cats kept in a house generally are spayed or neutered, so if we increase the concentration, will it be effective for them as well?

R: Even though in castrated cats the treatment solution did not reduce their interaction with the scratcher, the effect of the solution in reducing scratching behavior was significant. Therefore, it is still effective in castrated cats in terms of reducing starching behavior.   

  1. There is a possibility that presenting the scent of an unknown male animal could cause anxiety and stress to cats. I would say there is fear that this may induce other problematic behaviors. I would like to know how many cats exhibited stressful behaviors during the test.

R: thanks for the consideration. During the test, apparent stressful behaviors were not observed for both shelter and owned cats. This information is added to the discussion (L409-410). 

  1. The following points should also be added to the evolutionary considerations.

Is it possible that feline social rank and testosterone levels correlate with those smells? For example, are MMB and Butanoic Acid from males with higher testosterone levels more repellent? Please discuss this point.

R: Thanks for the insightful comment. The following content has been added to the discussion. “From evolutionary considerations, feline social rank and testosterone levels might correlate with the concentration of the semiochemicals in the urine and feces of intact males (Hendriks et al. 1995a; Miyazaki et al. 2006). Future studies might explore the correlation between these variables and test out the intraspecific effects of different combination of candidate volatiles from cat urine and feces in various concentrations. ” (L422-426)

Reviewer 2 Report

Comments and Suggestions for Authors

Thank you for inviting me to review this interesting and well-written manuscript. I have a few comments on the manuscript, I hope the authors find them helpful.

Comment1: Can the authors define the terms pheromones and semiochemicals? Throughout the manuscript the authors appear to use them interchangeably and it is also referred to as the “pedal interdigital pheromone” (L43) rather than as an interdigital semiochemical, as I have seen it other places (e.g., Cozzi et al., 2013). Are pheromones a form of semiochemical for communication within a species? Could the authors please clarify within the text what, if any, differences exist between the terms?

Cozzi et al. (2013) https://doi.org/10.1177/1098612X13479114 

Comment 2: In L39 is it both olfactory and chemical communication, or olfactory only?

Comment 3:
At L67, I also think it would be worth noting here (or maybe in the discussion) that the ability to engage in scratching behavior is not only natural, it is vital to the welfare of cats. One way to improve animal welfare by providing opportunities for that animal to engage in natural, species-typical behaviors.

Comment 4: At L165, was the introduction to the room part of the 20 mins or was there an additional habituation period prior to the start of the 20-min preference assessment?

Comment 5: At L166, please clarify why 20 min was chosen as the duration for the preference assessment. Was this based on other methodology or was there another reason? If based on other methods, please cite the source. 

Along with this, it does not seem like the cats interacted with the scratchers for very long (e.g., duration measures in Figure 4). For example, on the longer end, cats spent a total of 21.19 seconds scratching out of a possible 20 minutes. Could the authors comment on whether 20 minutes is actually necessary for the assessment? Could shorter assessments possibly be utilized in the future to make trials easier for both cats and researchers?

Comment 6: At L169 the authors mention the behavior coder was validated from intra-observer agreement in a prior study. Can the authors clarify why these methods were chosen? Was intra-observer agreement or inter-observer reliability calculated with data from this study? If not, is this common practice to use reliability scores from a different sample, but for the same behavior? In other behavioral studies I see IOR scores are presented for the data in the study itself. Are there any possible differences between the present study and the 2020 study that could cause differences in the reliability of these behavioral measures?

Comment 7: Is there a black cat in the center of Figure 1? If so, it is very hard to see because of the dark lighting. This isn't a big deal, but it would be nice if the authors could lighten the area around the cat so that it can be better seen.

Comment 8: At L375 the authors say “...because in free-roaming cats that usually live a solitary life during non-breeding season” Can the authors please clarify this sentence. Is it saying that free-roaming cats usually live solitarily and so we see the use of chemical communication? Or are you saying in free-roaming cats, who also live solitarily, we see the use of chemical communication? I question this because free-roaming cats do form highly gregarious social colonies with one another, and even these socially-living cats rely on chemical communication. So, this statement needs to be restated a bit for clarity. For examples, see papers on cat activity and social behavior in the recent free-roaming cat special issue https://www.mdpi.com/journal/animals/special_issues/free_ranging_cats 

Comment 9:
The authors provide a solid recommendation at L410. As I mention in my third comment, it might be worth noting how this type of research has the potential to inform recommendations that could increase cat welfare and improve human-cat relationships.

Author Response

Thank you for inviting me to review this interesting and well-written manuscript. I have a few comments on the manuscript, I hope the authors find them helpful.

  1. Can the authors define the terms pheromones and semiochemicals? Throughout the manuscript the authors appear to use them interchangeably and it is also referred to as the “pedal interdigital pheromone” (L43) rather than as an interdigital semiochemical, as I have seen it other places (e.g., Cozzi et al., 2013). Are pheromones a form of semiochemical for communication within a species? Could the authors please clarify within the text what, if any, differences exist between the terms?

Cozzi et al. (2013) https://doi.org/10.1177/1098612X13479114

R: Thanks for the comment, the highlighted content has been added to introduction. (L 41-47) And “pedal interdigital pheromone” has been replaced with “pedal interdigital semiochemicals”.

Semiochemicals include any chemical signal given off by one individual that provides a message, altering the physiology and/or behavior of another individual, while pheromones are a type of semiochemicals for communication within the same species (Vitale et al., 2018). In the current study, we will also use semiochemical to describe those chemical communicative compounds, the exact biological functions of which have not yet been completely understood.

  1. In L39 is it both olfactory and chemical communication, or olfactory only?

R:Thanks for the comment and “olfactory communications” has been replaced with “olfactory and chemical communications.” (L39)

  1. At L67, I also think it would be worth noting here (or maybe in the discussion) that the ability to engage in scratching behavior is not only natural, it is vital to the welfare of cats. One way to improve animal welfare by providing opportunities for that animal to engage in natural, species-typical behaviors.

R: we agree with the reviewer and have add the notion in introduction. (L71-73) “The ability to engage in scratching behavior is not only natural, but vital to the welfare of cats because one way to improve animal welfare is providing opportunities for that animal to engage in natural, species-typical behaviors.”

  1. At L165, was the introduction to the room part of the 20 mins or was there an additional habituation period prior to the start of the 20-min preference assessment?

R: there is no habituation period before the test. This information is added to methods. (L177)

  1. At L166, please clarify why 20 min was chosen as the duration for the preference assessment. Was this based on other methodology or was there another reason? If based on other methods, please cite the source.

R: The testing period of 20 minutes was not chosen based on other methodology. We chosed 20 minutes to allow extra time for the cat to explore and get used to the testing room, especially for shelter cats. 

“A habituation period was not specified, therefore the 20-minute testing period included the time for the cat to habituate to the room and explore both scratchers.” (L177-178)

  1. Along with this, it does not seem like the cats interacted with the scratchers for very long (e.g., duration measures in Figure 4). For example, on the longer end, cats spent a total of 21.19 seconds scratching out of a possible 20 minutes. Could the authors comment on whether 20 minutes is actually necessary for the assessment? Could shorter assessments possibly be utilized in the future to make trials easier for both cats and researchers?

R: Thanks for the comment. When we were coding the videos, we did realize that most cats finished interacting with the scratchers (including scratching) within the first 10 minutes of test period. We added the following recommendation in the discussion (L377-380). Besides, most cats in the current study finished interacting with both scratchers within the first 10 minutes of the testing period, therefore suggesting shorter assessments possibly be utilized in the future to make trials easier for both cats and researchers.

  1. At L169 the authors mention the behavior coder was validated from intra-observer agreement in a prior study. Can the authors clarify why these methods were chosen? Was intra-observer agreement or inter-observer reliability calculated with data from this study? If not, is this common practice to use reliability scores from a different sample, but for the same behavior? In other behavioral studies I see IOR scores are presented for the data in the study itself. Are there any possible differences between the present study and the 2020 study that could cause differences in the reliability of these behavioral measures?

R: We agree with the reviewer that the method we use for intra observer agreement in this study was not common. In fact, this current study and the 2020 study (Zhang et al., 2020) were content of different chapters of a PhD thesis from 2019, which was conducted and had video coded and data analyzed by the same researchers. Therefore, we suggest that the methodology in evaluating IOR adopted in the current study is valid. 

  1. Is there a black cat in the center of Figure 1? If so, it is very hard to see because of the dark lighting. This isn't a big deal, but it would be nice if the authors could lighten the area around the cat so that it can be better seen.

R: Thanks for the comment and we added label text next to the cat in the picture.

  1. At L375 the authors say “...because in free-roaming cats that usually live a solitary life during non-breeding season” Can the authors please clarify this sentence. Is it saying that free-roaming cats usually live solitarily and so we see the use of chemical communication? Or are you saying in free-roaming cats, who also live solitarily, we see the use of chemical communication? I question this because free-roaming cats do form highly gregarious social colonies with one another, and even these socially-living cats rely on chemical communication. So, this statement needs to be restated a bit for clarity. For examples, see papers on cat activity and social behavior in the recent free-roaming cat special issue https://www.mdpi.com/journal/animals/special_issues/free_ranging_cats

R: Thank you for the useful information and we have restated the sentence. (L388-389) “Alternatively, cats may avoid the treated-scratcher to avoid confrontation with other cats and a recent hunting area because free-roaming cats whether live a solitary life or form highly gregarious social colonies with one another (Jensen et al., 2022), use urinary and fecal marks to separate themselves apart temporally and spatially during hunting and social communication (Bradshaw et al. 2012).”

  1. The authors provide a solid recommendation at L410. As I mention in my third comment, it might be worth noting how this type of research has the potential to inform recommendations that could increase cat welfare and improve human-cat relationships.

R: We agree with the reviewer and “The mixed solution of MMB and butanoic acid may have an application in protecting furniture or other objects or areas from being scratched by cats, which has the potential to inform recommendations that could increase cat welfare and improve human-cat relationships.” has been added to conclusions. (L 435-436)

Reviewer 3 Report

Comments and Suggestions for Authors

This is an interesting and well-designed study of cat semiochemicals with potential practical applications.

Note Preference Index (PI) is first defined in Table 1 p. 5. The term PI is first used in the text p. 11, line 271. It should be redefined at this point for clarity.

 They note that MMB has been described in the literature as having typical male cat odor at .01 - 1 ppm, significantly lower than the MMB estimated in male and female cats in this study at 6.60 ppm and .81. (Line 310)

Given that the major potential practical application of this research is the development of an application to protect furniture and other objects from being scratched, it would be useful to provide additional information on the extent to which the necessary concentrations of such a solution would be detectable or even aversive to humans.

Author Response

This is an interesting and well-designed study of cat semiochemicals with potential practical applications.

  1. Note Preference Index (PI) is first defined in Table 1 p. 5. The term PI is first used in the text p. 11, line 271. It should be redefined at this point for clarity.

R: Thanks for pointing this out. Change has been made. (L286)

  1. They note that MMB has been described in the literature as having typical male cat odor at .01 - 1 ppm, significantly lower than the MMB estimated in male and female cats in this study at 6.60 ppm and .81. (Line 310)

R: Thanks for the comment. Change has been made. (L324-325)

  1. Given that the major potential practical application of this research is the development of an application to protect furniture and other objects from being scratched, it would be useful to provide additional information on the extent to which the necessary concentrations of such a solution would be detectable or even aversive to humans.

R: Thanks for the comment. We were not able to confirm the minimal concentration of treatment solution that can be unnoticeable or acceptable to humans while still exert deterring effect in the current study. This shortcoming is acknowledged and added to discussion. (L418-421).